# Soy Protein Isolate/Sodium Alginate Microparticles under Different pH Conditions: Formation Mechanism and Physicochemical Properties

**DOI:** 10.3390/foods11060790

**Published:** 2022-03-09

**Authors:** Jia Cao, Xiaohong Tong, Mengmeng Wang, Tian Tian, Sai Yang, Mingyue Sun, Bo Lyu, Xinru Cao, Huan Wang, Lianzhou Jiang

**Affiliations:** 1College of Food Science, Northeast Agricultural University, Harbin 150030, China; caojia1122163@163.com (J.C.); tongxiaohong0110@163.com (X.T.); wmm13206677625@163.com (M.W.); tiantian@neau.edu.cn (T.T.); s201001027@neau.edu.cn (S.Y.); neausmy@163.com (M.S.); michael_lvbo@163.com (B.L.); cxr19919@163.com (X.C.); 2Key Laboratory of Soybean Biology of Chinese Education Ministry, Harbin 150030, China; 3School of Tea and Food Science & Technology, Anhui Agricultural University, Hefei 230036, China

**Keywords:** soy protein isolate, sodium alginate, pH, microparticles, structure, physicochemical properties

## Abstract

The effects of sodium alginate (SA) and pH value on the formation, structural properties, microscopic morphology, and physicochemical properties of soybean protein isolate (SPI)/SA microparticles were investigated. The results of ζ-potential and free sulfhydryl (SH) content showed electrostatic interactions between SPI and SA, which promoted the conversion of free SH into disulfide bonds within the protein. The surface hydrophobicity, fluorescence spectra, and Fourier transform infrared spectroscopy data suggested that the secondary structure and microenvironment of the internal hydrophobic groups of the protein in the SPI/SA microparticles were changed. Compared with SPI microparticles, the surface of SPI/SA microparticles was smoother, the degree of collapse was reduced, and the thermal stability was improved. In addition, under the condition of pH 9.0, the average particle size of SPI/SA microparticles was only 15.92 ± 0.66 μm, and the distribution was uniform. Rheological tests indicated that SA significantly increased the apparent viscosity of SPI/SA microparticles at pH 9.0. The maximum protein solubility (67.32%), foaming ability (91.53 ± 1.12%), and emulsion activity (200.29 ± 3.38 m^2^/g) of SPI/SA microparticles occurred at pH 9.0. The application of SPI/SA microparticles as ingredients in high-protein foods is expected to be of great significance in the food industry.

## 1. Introduction

Soy protein isolate (SPI) has good nutritional value, can provide a proper balance of various essential amino acids for human health, and can be used as a good source of protein supplement for vegans [1]. As a nutritionally valuable food additive, SPI has attracted much attention. However, its physicochemical properties are affected by the surrounding environmental conditions compared to animal proteins, which limits its food processing applications [2]. The structure is an important factor in determining the quality of protein products [3]. Therefore, to improve the application of SPI in the food industry, it is necessary to modify its structure appropriately to improve its processing characteristics [4].

In recent years, the spatial structure of SPI has been changed through different methods to enhance its quality and processing characteristics. However, the traditional protein modification method led to negative effects on the physical and sensory attributes of the protein-enriched food [5]. Previous studies have indicated that the preparation of microparticles could improve the internal structural properties and microscopic morphology of the protein, and have favorable effects on the stability, emulsifying, and foaming properties [3,6]. However, structural and physicochemical properties of protein microparticles were closely related to the conditions of the surrounding environment during the microparticle process, and were mainly affected by the pH conditions [7]. The microparticulated whey proteins prepared by [8] in the low-acidic pH range exhibited substantially higher particle size reduction than at neutral pH, with improved solubility and emulsifying properties. However, zein microparticles prepared under various pH conditions markedly improved the viscosity of emulsions at alkaline conditions compared to acidic conditions [9]. The interaction between the zein and water molecules was stronger, which could overcome the strong hydrophobicity of zein under alkaline conditions [9]. The prepared zein microparticles were more uniform and smaller, with better hydration performance.

Protein/polysaccharide combinations have found widespread applications because of their superior physicochemical properties (such as rheological properties, dispersibility and emulsifying properties, etc.) to the pure protein [10]. Moreover, when prepared from food-grade ingredients, they are usually used as additives to enhance the structure and stability of food systems [11]. Complexation with polysaccharides can improve the viscosity, solubility, and emulsifying properties [11]. However, the formation of complexes requires careful consideration of the compatibility between the selected protein and polysaccharide and their sensitivity to chemical, physical, and structural parameters [12]. Some of the important parameters are the pH value, ionic strength, and biopolymer weight ratio [13,14].

Sodium alginate (SA) is a linear polyanionic natural polysaccharide widely used as a stabilizer, solubilizer, and emulsifier in food systems [15] due to its numerous hydrophilic carboxyl and hydroxyl groups and unique colloidal properties [16]. Owing to the linear structure of SA, it can form a more ordered network structure with protein, reducing the random binding of protein in the system [17]. Li et al. [14] obtained a soluble form of myofibrillar proteins under low-salt conditions by mixing with SA. When the myofibrillar protein/SA mixing ratio was 20 and 5, the precipitation of myofibrillar proteins was completely inhibited. Compared with pure protein, the soluble complex formed by SA and protein increased foam stability, and its foam half-life could be increased by two times [18]. Furthermore, SPI/SA complexes showed improved emulsion stability compared to SPI [19].

To date, few studies have been reported on microparticles prepared from SPI and polysaccharides. Furthermore, the effect of pH on the interaction mechanism and physicochemical properties of proteins and polysaccharides during the formation of microparticles need to be further clarified. In this study, SPI was used as the raw material, and SPI/SA microparticles were prepared by SA modification and pH control. The formation mechanism, structural properties, particle morphology, and physicochemical properties of SPI/SA microparticles were studied. The results provide the theoretical foundation and technical support for the application of SPI/SA microparticles in food systems as a new generation of functional food ingredients.

## 2. Materials and Methods

### 2.1. Materials

Soybean protein isolate (SPI, 93.8% protein) was prepared by the reported method [20]. Protein content was determined by the Dumas method (N × 6.25) in a nitrogen/protein analyzer (Rapid N Cube, Elementar Analysensysteme GmbH, Hanau, Germany). Corn oil was purchased from a local supermarket (Harbin, China). Sodium alginate (SA), 8-anilino-1-naphthalenesulfonic acid (ANS), and a BCA protein assay kit were bought from Sigma-Aldrich Chemical Co., Ltd. (St. Louis, MO, USA). All other reagents were purchased from local shops in China.

### 2.2. Preparation of SPI and SPI/SA Microparticles

SPI and SA were dissolved in deionized water at concentrations of 4 wt% and 1 wt%, respectively, by stirring for 2 h at 25 °C. SPI/SA (1:1, *v*/*v*) solutions were mixed and stirred for 30 min, and the pH was adjusted to 3.0, 5.0, 7.0, 9.0, and 11.0. The mixtures were placed in a water bath for 30 min at 80 °C, and then sheared using a high-speed homogenizer at 12,000 rpm for 3 min and, finally, passed through a high-pressure homogenizer twice at 30 MPa, and its pH adjusted back to 7.0. These mixtures were spray-dried by a mini spray dryer to obtain SPI/SA microparticles. The spray-drying conditions were as follows: inlet temperature, 180 °C; outlet temperature, 90–110 °C; injection flow rate, 3 mL/min; air flow rate, 600 L/h. The spray-dried SPI/SA microparticles were named SPI/SA_pH3_, SPI/SA_pH5_, SPI/SA_pH7_, SPI/SA_pH9_, and SPI/SA_pH11_, respectively. SPI microparticles were prepared as the controls, and named SPI_pH3_, SPI_pH5_, SPI_pH7_ SPI_pH9_, and SPI_pH11_, respectively [5].

### 2.3. ζ-Potential Measurement

The ζ-potential of the samples was measured using a zeta potential analyzer (Zeta Plus, Malvern, UK) at 25 °C. The refractive indices of the sample and dispersed medium were 1.52 and 1.33, respectively [21].

### 2.4. Surface Hydrophobicity (H_0_) Measurement

The microparticles were diluted to a protein concentration of 0.001–0.2 mg/mL with deionized water; then, 20 μL of 10 mM ANS solution was added to 2 mL of the sample solutions. The fluorescence intensity of the samples was measured by a fluorescence spectrophotometer (Hitachi Co., Tokyo, Japan) at the excitation wavelength of 390 nm, emission wavelength of 470 nm, and a slit width of 5 nm. The *H*_0_ was calculated from the relative fluorescence intensity, and the initial slope of protein concentration by linear regression analysis [22].

### 2.5. Free SH Content Measurement

Based on a previous report, and with some modifications, the free sulfhydryl (SH) content in SPI and SPI/SA microparticles was determined [5]. Then, 2 mL of the sample solution with 1 wt% protein content was added to 10 mL of Tris–glycine urea buffer with 20 μL of Ellman’s reagent, then mixed for 15 min. The absorbance was measured at 412 nm after mixing for 15 min at 25 °C. Free SH content was calculated as follows:−SH (μmol/L) = (75.35 × *A*_421_)/*C*(1)
where 73.53 is the molar extinction coefficient of Ellman’s reagent; *A*_412_ is the absorbance value at 412 nm; *C* is the protein concentration (mg/mL).

### 2.6. Intrinsic Fluorescence Spectra

The intrinsic fluorescence spectra were measured using a fluorescence spectrophotometer at an excitation wavelength of 295 nm and an emission wavelength from 300 to 400 nm, and the slit width was 5 nm. The samples were diluted to a protein concentration of 0.01 mg/mL with deionized water [23].

### 2.7. Fourier Transform Infrared Spectroscopy (FTIR)

The samples were mixed evenly with potassium bromide (KBr) at 1:100 (*w*/*w*), pressed into a tablet, and then placed in the sample holder of an FTIR spectrometer. The spectral scanning range was 4000–400 cm^−1^ with a resolution of 4 cm^−1^, and 64 scans per sample. PeakFit Version 4.12 software was used to calculate the change in secondary structure content within the amide I region (1700–1600 cm^−^^1^) [24].

### 2.8. Differential Scanning Calorimetry (DSC) Analysis

The thermal properties of the samples were characterized using a differential scanning calorimeter (DSC 8000, PE Instruments, Inc., Waltham, MA, USA). The sample (2.00 mg) was placed in an aluminum pot, and sealed with an aluminum lid. The initial scanning temperature was set from 30 to 180 °C at a constant rate of 10 °C/min, and an empty aluminum pan served as the reference [25].

### 2.9. Particle Size Measurement

Particle size was determined using Malvern MasterSizer 3000 (Malvern Instruments Ltd., Worcestershire, UK) at 25 °C. The refractive indices of the sample and the dispersed medium were 1.52 and 1.33, respectively [26].

### 2.10. Scanning Electron Microscopy (SEM)

The morphology of samples was observed using a benchtop scanning electron microscope. The samples were glued with a conductive adhesive to a stainless-steel stage. It was coated with gold in an E-1010 ion sputter, and then observed at an accelerating voltage of 15.0 kV [27].

### 2.11. Apparent Viscosity Measurement

The samples were diluted to a protein concentration of 10 wt%. Steady shear viscosity of the dispersion was characterized at 25 °C by a rotational rheometer with parallel plates (diameter of 40 mm) and a 0.5 mm measurement gap over the shear rate range of 0.1–100 s^−1^ [28].

### 2.12. Protein Solubility Measurement

The samples were diluted to a protein concentration of 1 wt%, stirred for 2 h at 25 °C, and then centrifuged at 10,000 rpm for 20 min. The protein content of the supernatant was determined by the bicinchoninic acid (BCA) protein assay kit. The protein solubility was expressed as the protein content in the supernatant compared to the total protein content [29].

### 2.13. Foaming Capacity (FC) and Foaming Stability (FS) Measurement

The FC and FS determination methods of [30] were slightly adjusted in this study. The sample was diluted to a protein concentration of 1 wt%. Then, 50 mL of the dispersion was homogenized using a high-speed homogenizer at 10,000 rpm for 2 min. The volume of foam after homogenization was measured at 0- and 30-min. FC (m^2^/g) and FS (%) were calculated as follows:FC (%) = (*V*_0_ − *V*)/*V* × 100(2)
FS (%) = (*V_t_*/*V*)/(*V*_0_ − *V*) × 100 (3)
where *V* is the volume of solution before homogenization; *V*_0_ and *V_t_* are the volumes of foam after homogenization at 0 and 30 min, respectively.

### 2.14. Emulsifying Activity Index (EAI) and Emulsifying Stability Index (ESI) Measurement

The sample was dispersed with 40 mL of deionized water to a protein concentration of 1 wt%. The dispersion was homogenized with 10 mL of corn oil using a high-speed homogenizer at 12,000 rpm for 2 min to obtain the emulsion. Then, 80 µL of the emulsion was added to 8 mL of 0.1 wt% SDS. The absorbance was measured at 500 nm at 0 and 30 min. EAI (m^2^/g) and ESI (%) were calculated as follows:EAI = (2 × 2.303 × *A*_0_ × *N*)/(*ρ* × *φ* × 100)(4)
ESI = *A*_0_/(*A*_0_ − *A*_30_) × 30(5)
where *N* is the dilution factor; *ρ* is the mass concentration of protein before the formation of the mixture (g/mL); *φ* is the volume fraction of the oil phase in the emulsion (%); *A*_0_ and *A*_30_ are the absorbances at 0 and 30 min, respectively [31].

### 2.15. Statistical Analysis

All measurements were carried out at least in triplicate. Significant differences (*p* < 0.05) were calculated by one-way analysis of variance (ANOVA) and Duncan’s multiple range test using IBM SPSS Statistics 19. Origin 2021 (Origin Lab, Northampton, MA, USA) was used for data analysis.

## 3. Results and Discussion

### 3.1. ζ-Potential Analysis

The ζ-potential is related to the surface charge of the molecule, and is an important indicator to evaluate the interaction mode and stability of the system [11]. Figure 1A shows the ζ-potential of SPI and SPI/SA microparticles. When the pH was 3.0, the SPI microparticles exhibited a positive charge due to protonation of the amino group at pH below the isoelectric point (pI) of the protein (pH = 4.5) [10]. At pH 5.0–11.0, the ζ-potential of the SPI microparticles changed from positive to negative charge. As the pH continued to increase, the absolute value of the ζ-potential of the SPI microparticles increased, which was attributed to the increased ionization of the carboxyl groups of the protein [32]. By contrast to the SPI microparticles, the SPI/SA microparticles all demonstrated a negative ζ-potential. At pH 3.0, the ζ-potential value was close to 0, which indicated that the electrostatic attraction between SPI and SA occurred when the pH was lower than the pI [33]. At pH 5.0–11.0, the ζ-potential value increased from −18.40 ± 0.46 to 39.47 ± 47 mV because SA hindered the aggregation of SPI, and exposed more charged amino acid residues [16]. Generally, the system was more stable when the absolute value of the ζ-potential was greater than 20 mV [22]. Furthermore, the highly branched neutral sugar side chains on SA stabilized the structure, and inhibited the aggregation of the complex [34].

### 3.2. Surface Hydrophobicity (H_0_) Analysis

The *H*_0_ plays a vital role in the aggregation of protein molecules, and the interface properties of the protein/polysaccharide [35]. The ANS fluorescent probe was used to measure the *H*_0_ of SPI and SPI/SA microparticles under different pH conditions. From Figure 1B, the *H*_0_ of SPI microparticles decreased as the pH increased from 3.0 to 5.0. However, when the pH increased from 7.0 to 11.0, the *H*_0_ gradually increased. Due to the unfolding of the protein structure, some of the hydrophobic groups buried inside the protein were exposed [36]. The *H*_0_ of the SPI/SA microparticles was lower than that of the SPI microparticles at the same pH value. It indicated that SPI and SA had a hydrophobic interaction, with partial coverage of the hydrophobic groups exposed on the surface of the protein [37].

### 3.3. Free SH Content Analysis

The free SH groups, disulfide bonds, and their interactions affect the structural properties of most proteins [38]. As depicted in Figure 1C, except at pH 5.0, the free SH content of the SPI microparticles did not change significantly under different pH conditions. However, the free SH content of the SPI/SA microparticles was significantly lower than that of the SPI microparticles, indicating that conformational changes induced by the addition of SA triggered the formation of new disulfide bonds due to increased reactivity between free SH groups [5]. In a previous study, the formation of disulfide bonds decreased the distance between amino acid residues in different regions of the same or different peptide chains [19]. The peptide chain folded quickly, and formed a stable spatial structure, and a certain amount of disulfide bonds was conducive to stabilizing the structure of microparticles [5].

As the pH increased from 3.0 to 9.0, the free SH content of the SPI/SA microparticles increased from 3.24 to 4.16 μmol/g. This change might be due to a change in the secondary and tertiary structure of the protein, causing part of the protein structure to unfold so that some free SH groups were exposed to the molecular surface [39]. However, when the pH was further increased to 11.0, the free SH content of the SPI/SA microparticles increased by only 0.16 μmol/g. As the pH increased from 9.0 to 11.0, the free SH content changed very little, so the subsequent experiments in this study were not carried out at pH 11.0.

### 3.4. Fluorescence Spectra Analysis

The fluorescence spectra of SPI and SPI/SA microparticles are shown in Figure 2A. When the pH increased from 3.0 to 9.0, the maximum fluorescence intensity of the SPI microparticles first increased and then decreased. At pH 5.0, the maximum fluorescence intensity decreased to the lowest. This might be due to hydrophobic-interaction-induced aggregation of the protein at a pH near the pI [32]. At a pH far away from the pI, the unfolding of the protein structure caused more Trp and Tyr residues to be exposed, so the maximum fluorescence intensity of the SPI microparticles increased [27]. The fluorescence intensity of the SPI/SA microparticles was lower than SPI microparticles at the same pH, indicating that SA quenched the intrinsic fluorescence of SPI. The hydrophobic interaction between SPI and SA blocked the exposure of hydrophobic amino acid residues (e.g., Trp), which was consistent with the reported combination of SPI and oxidized bacterial cellulose [37]. In addition, there was an interaction between the polar regions of SA and the SPI, whereas the non-polar regions formed a hydrophobic environment of Trp residues inside the microparticles, which led to a decrease in the maximum emission wavelength and fluorescence intensity of the SPI/SA microparticles [35].

### 3.5. FTIR Spectra Analysis

FTIR is often used to characterize molecular interactions, and analyze protein secondary structures [12]. The changes in some side groups and microenvironments of protein obtained by analyzing the FTIR bands allow determining the formation of new compounds or functional groups, and the types of intermolecular forces [40]. The FTIR spectra of SPI and SPI/SA microparticles are shown in Figure 2B. The absorption peak of the SPI/SA microparticles shifted at 3000–3700 cm^−1^, and a noticeably wider band appeared, which indicated that SPI and SA interacted to form hydrogen bonds [10]. The characteristic peak at 2960.97 cm^−1^ corresponded to the change in the C-H tensile vibration due to the hydrophobic interaction between SPI and SA [41]. The characteristic peak of the SPI microparticles at 1655.31 cm^−1^ was due to the stretching vibration of the amide I zone with C=O; the absorption peak at 1535.56 cm^−1^ was the C-N stretch and N-H in the amide II zone; and the vibrational absorption peaks of -CH and -CH_3_ appeared at 1450.55 and 1397.07 cm^−1^, respectively [42]. Relative to the absorption peaks of SPI microparticles, new absorption peaks were observed at 1539.38 and 1400.36 cm^−1^ for the SPI/SA microparticles. It showed that the -NH_3_^+^ groups on SPI and the -COO^−^ groups on SA had an electrostatic attraction [43]. Thus, the formation of SPI/SA microparticles was the result of hydrogen bonds, hydrophobic interactions, and electrostatic interactions.

The influence of pH on the protein secondary structure in SPI and SPI/SA microparticles is shown in Figure 2C. Compared with SPI microparticles, the content of α-helix and random coils of SPI/SA microparticles were reduced, the content of β-sheets was increased, and the content of β-turns did not change significantly. These results indicated that the addition of SA changed the conformation of the protein, and promoted the transition from a disordered to an ordered structure [32]. In addition, the *H*_0_ of SPI/SA microparticles was lower than that of the SPI microparticles because the hydrophobic interaction between SA and SPI reduced the formation of disordered structures [41]. With the increase of pH from 3.0 to 9.0, the β-sheet content of the SPI/SA microparticles gradually increased from 32.24% to 37.17%. Therefore, SA induced the formation of ordered structures in SPI/SA particles at pH 9.0.

### 3.6. DSC Analysis

The thermal stability of the protein in food determines the processing conditions of the food, and an improvement in the thermal stability can increase the processing range of the protein [44]. Figure 3A,B display the thermal stability of SPI and SPI/SA microparticles under different pH conditions. The maximum denaturation temperature of the SPI/SA microparticles (99.82 °C) was significantly higher than that of the SPI microparticles (93.82 °C), indicating that after combining with SA, the hydrophobic and electrostatic interactions in the molecule increased the stability of the protein structure, and the thermal stability [45]. When considered in conjunction with the spectroscopy data, these findings indicate that pH could affect the conformation of SPI and SPI/SA microparticles, thereby affecting their thermal stability. Jiang et al. [25] confirmed that the heat denaturation temperature of the mesona chinensis polysaccharide/whey protein isolate complex was higher than that of whey protein isolate, indicating that polysaccharides could increase the thermal stability of the protein. The results of Lee et al. [45] showed that chitosan could induce α-lactalbumin to form a stable structure, consequently improving its thermal denaturation temperature. Therefore, the regulation of pH and the addition of SA could modify the structure of the protein, in turn, improving its thermal stability via the formation of SPI/SA microparticles.

### 3.7. Particle Size Analysis

Results of the average particle size analysis of SPI and SPI/SA microparticles are presented in Table 1. As the pH increased, the average particle size of the SPI microparticles increased and then decreased. The average particle size had a maximum value (38.32 ± 0.72 μm) at pH 5.0, and the minimum size (13.89 ± 0.92 μm) occurred at pH 9.0 [5]. Compared with the SPI microparticles, except at pH 5.0, the average particle size of the SPI/SA microparticles gradually decreased at pH 3.0–9.0. It could be that the SPI microparticles formed in the spray-drying process shrink and become collapsed spheres, resulting in a decrease in particle size, whereas the SPI/SA microparticles maintained their full spherical shape due to a solid interior [37]. It is worth noting that at pH 5.0, the average particle size of the SPI/SA microparticles (28.04 ± 0.89 μm) was smaller than that of the SPI microparticles (38.32 ± 0.72 μm), indicating that the addition of SA helped to inhibit the aggregation of SPI at a pH near the pI, and synergistically stabilize the interfacial film of the microparticles [7].

The particle size distribution curve of the SPI and SPI/SA microparticles are shown in Figure 3C,D, respectively. When the pH increased from 3.0 to 5.0, the particle size distribution curve of the SPI microparticles changed from a single peak to a double peak, and the particle size increased (Figure 3C). Moreover, as the pH increased from 7.0 to 9.0, the particle distribution curve shifted to the left. The particle size distribution curves of all SPI/SA microparticles, except at pH 5.0, shifted to the right (Figure 3D), which was consistent with the average particle size results above. It indicated that the complex formed by SA and SPI through electrostatic attraction and hydrophobic interactions contributed to the formation of a thick interfacial film [32]. In addition, the disulfide bonds formed upon the complexation of SA and SPI were conducive to forming SPI/SA microparticles with a compact internal space, endowing a certain rigid structure that was not prone to collapse [39]. At pH 3.0–9.0, the particle size distribution curve of the SPI/SA microparticles gradually moved to the left; that is, the particle size gradually decreased. The particle size range of the SPI/SA microparticles was the smallest at pH 9.0, and the average particle size was only 14.83 ± 0.83 μm. This indicated that SA improved the flexibility of SPI at a pH far away from the pI, thereby forming SPI/SA microparticles with a relatively smaller average particle size, and more uniform distribution [11].

### 3.8. Morphology Analysis

The pH conditions have an important effect on the aggregation behavior of proteins, and the morphology of the microparticles formed [36]. Figure 4 shows the effect of different pH values on the microstructure of the SPI and SPI/SA microparticles. At pH 3.0, the spray-dried SPI microparticles displayed a rough and cracked surface because the ordered structure of the proteins is destroyed under acidic conditions, and the protein film on the surface of the droplets is relatively thin and uneven [32]. During spray-drying, when the atomized droplets conduct heat and mass transfer with hot air in the drying chamber, the protein molecules migrate to the surface of the droplets along with the moisture, causing the SPI microparticles to rupture [5].

The SPI microparticles were irregular solid spheres at pH 5.0 because at a pH close to the pI, the repulsion between protein molecules is reduced, and large aggregates are formed [46]. As the pH increased from 7.0 to 9.0, the SPI microparticles gradually became collapsed hollow spheres because at a pH far away from the pI, the structure of the protein unfolded [6]. After the moisture evaporated, the protein film was not strong enough to maintain its spherical shape, and collapsed [6].

In comparison to the SPI microparticles, the SPI/SA microparticles displayed a fuller appearance. SA enabled SPI to migrate to the droplet surface to form a more uniform and thicker protein film during spray-drying [47]. As the pH increased from 7.0 to 9.0, the SPI/SA microparticles gradually became closer to a regular spherical shape, and the surface was smoother. The presence of SA during spray-drying led to a more compact structure of SPI, which enhanced the strength of the protein film on the droplet surface [48]. Especially, it was found that pH 9.0 was most conducive to forming SPI/SA microparticles with a uniform size. Under alkaline conditions, the polypeptide chains in the protein stretch. As a result, the protein occupied more volume in the suspension after SA than at other pH conditions, so the ordered structure of the complex formed by the SPI and SA through partial disulfide bonds or hydrogen bonds contributed to the formation of a uniform and dense protein film, leading to the formation of more uniform, spherical SPI/SA microparticles [37].

### 3.9. Apparent Viscosity Analysis

The curves of the apparent viscosity of the SPI and SPI/SA microparticles as a function of shear rate are shown in Figure 5A,B. Apparent viscosity decreased with the increase of shear rate, indicating that both SPI and SPI/SA microparticles were pseudoplastic fluids with shear-thinning characteristics [5]. The apparent viscosity tended to be stable at the shear rate of 40 s^−1^. Therefore, we evaluated the apparent viscosity of the samples at the shear rate of 40 s^−1^. In Figure 5A, the apparent viscosity of the SPI microparticles gradually decreases at pH 3.0–5.0 and then increases at pH 7.0–9.0. When the pH was 5.0, the apparent viscosity of the SPI microparticles was lowest (0.016 Pa·s). Due to the aggregation of protein at pH near the pI because of the decrease in electrostatic repulsion, a large number of hydrophobic groups and hydrogen bonding sites are hidden, resulting in the weakening of the interaction between the protein; therefore, the apparent viscosity of the SPI microparticles was reduced [28]. The apparent viscosity of the SPI/SA microparticles was considerably higher than SPI microparticles at the same pH, especially when the pH was 9.0 (0.85 Pa·s) (Figure 5B). The molecular weight of the complex formed by SPI and SA was markedly higher than that of SPI, and the hydrophilic groups on the molecular chain were fully hydrated and stretched to form a network structure that could hinder external shear [49].

### 3.10. Protein Solubility Analysis

The protein solubility of the SPI and SPI/SA microparticles is illustrated in Figure 5C. With the increase of pH, the protein solubility of the SPI microparticles decreased first and then increased, with a minimum value at pH 5.0. As the net charge of protein molecules at a pH near the pI is close to 0, aggregation and precipitation occur, which causes the protein solubility of SPI microparticles to decrease [32]. At pH 7.0–9.0, the protein solubility of the SPI microparticles gradually increased. Perhaps the increase in the electrostatic repulsion between protein reduced its aggregation tendency, which increased the relative surface area of the SPI microparticles, and the protein solubility. In addition, the protein solubility was significantly improved when SA was added because the negative charge of the SPI/SA microparticles was increased, which reduced the binding force between protein molecules [36]. At pH 3.0–9.0, the protein solubility of the SPI/SA microparticles gradually increased from 34.24% to 67.32%. As the structure of the complex formed by SPI and SA became more stretched, part of the polar groups inside the protein were exposed, which promoted the hydration of the protein, and improved the protein solubility of the SPI/SA microparticles [50].

### 3.11. FC and FS Analysis

Protein molecules are amphiphilic in nature and, as a result, are highly surface-active [51]. The foaming properties are usually characterized by FC and FS [52]. Figure 6A,B illustrate the FC and FS of the SPI and SPI/SA microparticles. With the increase of pH, the FC and FS of the SPI microparticles first decreased and then increased, reaching the lowest value of 53.9.32 ± 0.66% and 64.40 ± 1.06%, respectively, at pH 5.0. Likely, the low solubility of the protein at a pH near the pI (4.5) led to insufficient protein at the air/water interface [53]. Compared to the SPI microparticles, the FC and FS of the SPI/SA microparticles were improved to varying degrees at the same pH value. Especially when the pH was 9.0, the SPI/SA microparticles showed the best FC (91.53 ± 1.12%) and FS (91.58 ± 0.92%). The combination of SPI and SA was beneficial for the formation of an ordered protein structure during foam formation, which improved the flexibility of the protein structure, and the adsorption rate at the water/air interface [52]. In addition, the complex formed by SPI and SA produced a dense and stable foam by effectively reducing the drainage rate [51]. Therefore, an appropriate pH and SA could improve the foaming performance of SPI/SA microparticles.

### 3.12. EAI and ESI Analysis

The EAI and ESI of SPI and SPI/SA microparticles are shown in Figure 6C,D. With the increase of pH, the EAI and ESI of SPI microparticles increased first and then decreased (*p* < 0.05). After adding SA, the EAI and ESI were significantly improved. The addition of SA improved the interfacial adsorption of SPI such that SPI could quickly adsorb at the oil/water interface, which improved the EAI of the SPI/SA particles [35]. Furthermore, SA increased the viscosity of the water phase in the oil/water emulsion system, and, at the same time, reduced the oil/water interfacial tension so that the ESI was improved [54]. As the pH increased from 3.0 to 9.0, the EAI and ESI of the SPI microparticles were gradually increased. The SPI/SA microparticles showed the best EAI and ESI of 200.29 ± 3.38 m^2^/g and 2353.43 ± 45.87%, respectively, at pH 9.0. It could be that the interaction between SPI and SA partially unfolded the protein molecular structure [35]. The exposed hydrophobic groups were conducive to the adsorption of the protein at the oil/water interface, thereby improving the EAI of the SPI/SA microparticles [35]. Moreover, the increased electrostatic repulsion of the SPI/SA microparticles was also beneficial for improving the emulsifying stability [32].

## 4. Conclusions

In summary, the interaction between SPI and SA for the formation of SPI/SA microparticles had important effects. The pH value had a significant influence on the conformation and physicochemical properties of the SPI/SA microparticles. The electrostatic and hydrophobic interactions between SPI and SA promoted the conversion of free SH groups in the protein to intermolecular disulfide bonds. The hydrogen bonding between SPI and SA in the SPI/SA microparticles was conducive to the formation of a more ordered, rigid structure, and improved thermal stability. In addition, when the pH was 9.0, the high electrostatic repulsion maintained the internal microenvironment of the SPI/SA composite, forming uniformly distributed microparticles with a smooth surface. The SPI/SA microparticles displayed significantly improved apparent viscosity, protein solubility, foaming, and emulsifying properties compared to the SPI microparticles. By adding SA, and controlling the pH value, SPI/SA microparticles with good characteristics could be developed, which provides an important reference for the wider application of SPI. Therefore, the application of SPI/SA microparticles as a solubilizer, emulsifier, and stabilizer in food systems (such as beverages, ice cream, and yogurt, etc.) is of great significance. In addition, in order to further clarify the application range of the microparticles, further studies on their flavor, mouthfeel, and digestive properties are required in the future.

## Figures and Tables

**Figure 1 foods-11-00790-f001:**
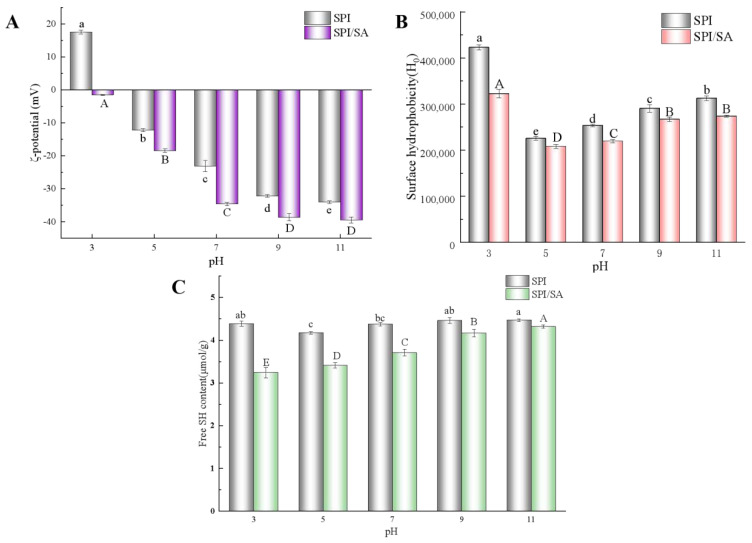
The ζ-potential (**A**), surface hydrophobicity (*H*_0_) (**B**), and free SH content (**C**) of SPI and SPI/SA microparticles at different pH values. Different lowercase letters (a–e) within a column for each SPI microparticles are significantly different (*p* < 0.05). Different capital letters (A–E) within a column for each SPI/SA microparticles are significantly different (*p* < 0.05).

**Figure 2 foods-11-00790-f002:**
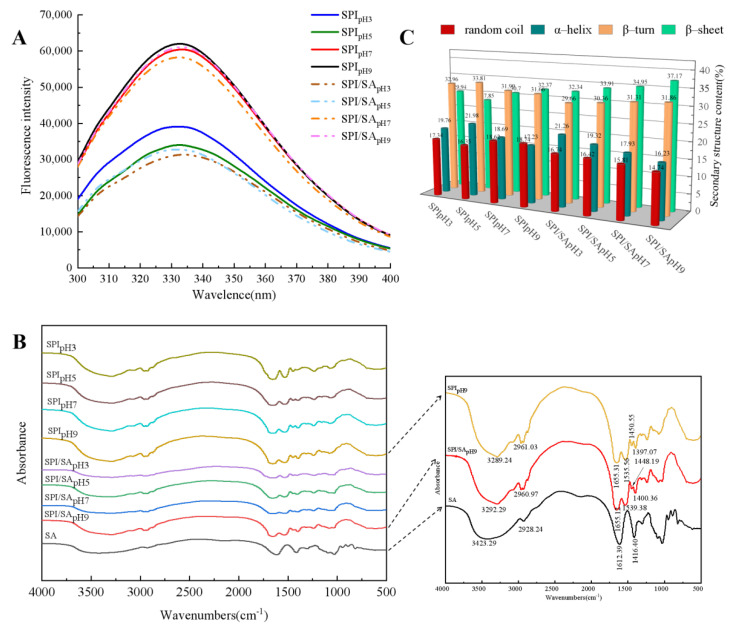
Fluorescence spectra (**A**), FTIR spectra (**B**), and secondary structure content (**C**) of SPI and SPI/SA microparticles at different pH values.

**Figure 3 foods-11-00790-f003:**
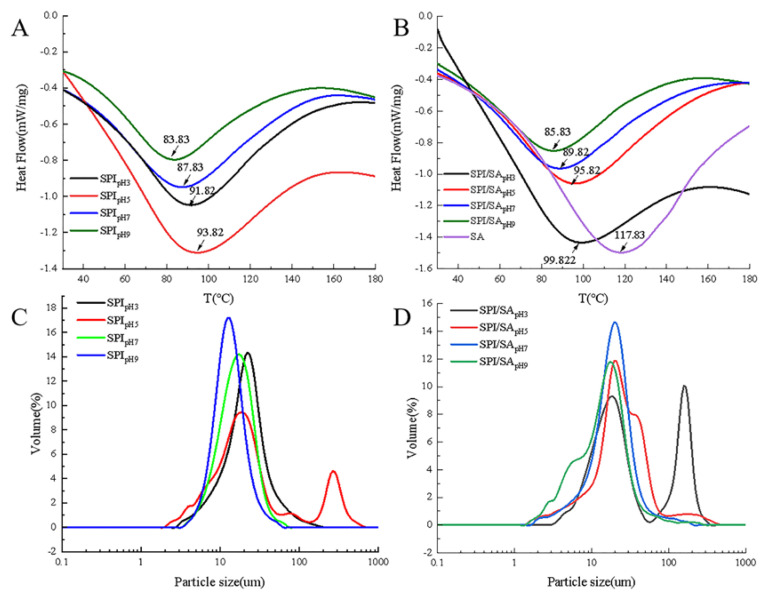
Thermal stability (**A**,**B**) and particle size (**C**,**D**) of SPI and SPI/SA microparticles at different pH values.

**Figure 4 foods-11-00790-f004:**
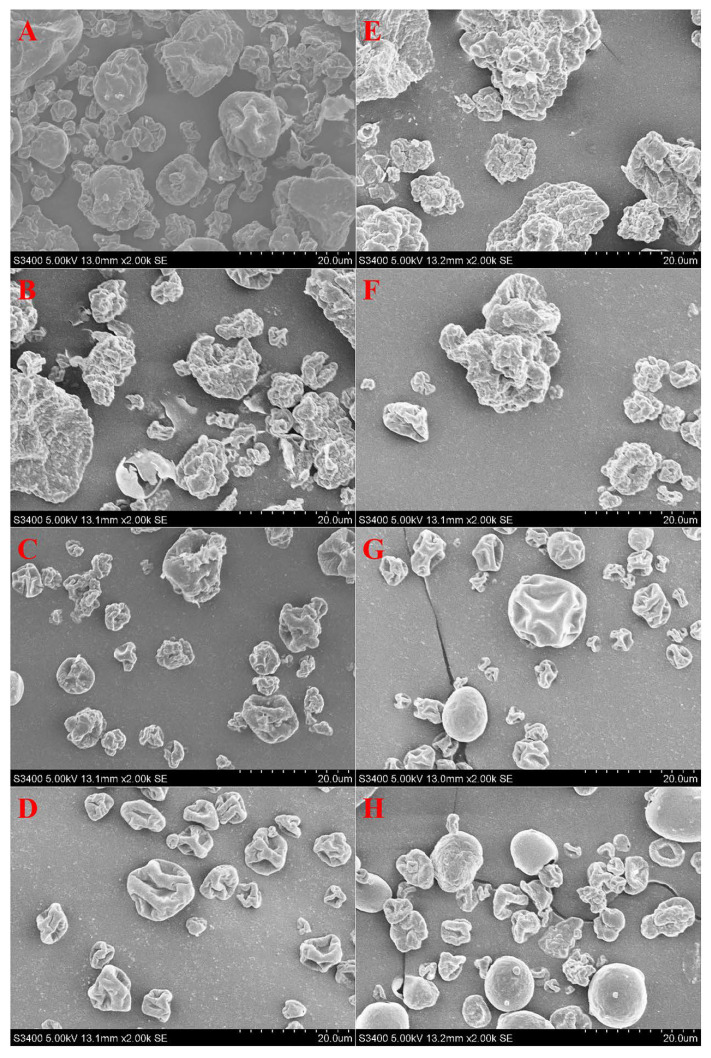
SEM images of SPI_pH3_ (**A**), SPI_pH5_ (**B**), SPI_pH7_ (**C**), SPI_pH9_ (**D**), SPI/SA_pH3_ (**E**), SPI/SA_pH5_ (**F**), SPI/SA_pH7_ (**G**), and SPI/SA_pH9_ (**H**) at different pH values.

**Figure 5 foods-11-00790-f005:**
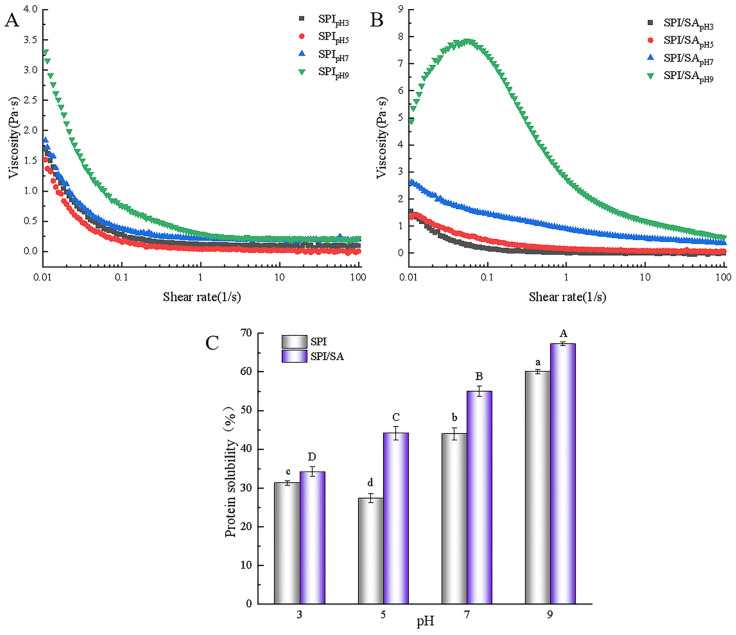
Apparent viscosity (**A**,**B**) and protein solubility (**C**) of SPI and SPI/SA microparticles at different pH values. Different lowercase letters (a–d) within a column for each SPI microparticles are significantly different (*p* < 0.05). Different capital letters (A–D) within a column for each SPI/SA microparticles are significantly different (*p* < 0.05).

**Figure 6 foods-11-00790-f006:**
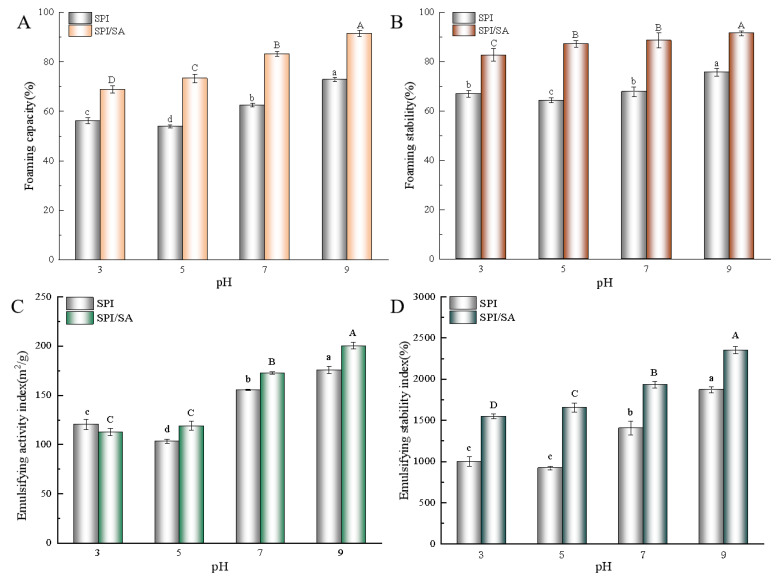
Foaming capacity (FC) (**A**), foaming stability (FS) (**B**), emulsifying activity index (EAI) (**C**), and emulsifying stability index (ESI) (**D**) of SPI and SPI/SA microparticles at different pH values. Different lowercase letters (a–d) within a column for each SPI/SA microparticles are significantly different (*p* < 0.05). Different capital letters (A–D) within a column for each SPI/SA microparticles are significantly different (*p* < 0.05).

**Table 1 foods-11-00790-t001:** The average particle size of SPI and SPI/SA microparticles at different pH values.

pH	SPI	SPI/SA
	Average Particle Size (μm)	Average Particle Size (μm)
3.0	26.52 ± 1.15 ^b^	63.40 ± 1.61 ^A^
5.0	40.39 ± 1.78 ^a^	36.89 ± 0.72 ^B^
7.0	18.42 ± 0.76 ^c^	24.00 ± 0.59 ^C^
9.0	14.77 ± 0.85 ^d^	15.92 ± 0.66 ^D^

The values are means ± standard deviation. The values are means ± standard deviation. Different lowercase letters (a–d) within a column for each SPI microparticles are significantly different (*p* < 0.05). Different capital letters (A–D) within a column for each SPI/SA microparticles are significantly different (*p* < 0.05).

## Data Availability

Not applicable.

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
