# Peer review of "Soy Protein Isolate/Sodium Alginate Microparticles under Different pH Conditions: Formation Mechanism and Physicochemical Properties"

_foods, 2022, doi:10.3390/foods11060790_

Round 1
Reviewer 1 Report
Some corrections are marked in PDF require.

Author Response
We thank the reviewer for comments, and contribution to the improvements to this manuscript. We have carefully revised, supplemented, and explained in accordance with the reviewers' comments and questions, and checked and checked the language of the full text. According to the reviewer’s comments, our responses are as follows in word.

Reviewer 2 Report
The paper penned by Cao et al is interesting and reader friendly. The manuscript is written in good English and the author makes reference to many relevant references. The article has been written very clear and well organized. The work is of interest and represents a very useful contribution to increase of knowledge in this field. I RECOMMEND MINOR REVISION.
Some comments to improve the article:
please keep an eye on the font size in your work
Title
I would change it, because you mention that you are studying functional properties, which include the most important one, gelation, and….. this is not included in the paper. of course I am not suggesting that you study gelation, just modify the title because the current one is misleading.
Line 39 – “Soy protein isolate (SPI) is a high-quality plant protein with a nutritional value comparable to animal protein” - I disagree with this statement, please rephrase this sentence, plant-based protein is simply inferior. Period. Please, write something about the SPI as a foaming alternative for vegans and it will be fine.
Line 65 -…. of their superior functional properties to the pure protein – something is missing in this sentence. Did you mean? “…of their superior functional properties compared to the pure protein.”?
Materials and methods
Despite the literature reference, it would be useful to add how much protein was exactly in SPI, what method was this content tested by?, or the manufacturer's declaration?
How were the SPI solutions prepared, basing on the pure protein calculation?
No information on sodium alginate
Results and discussion
Zeta potential paragraph – please cite the following paper:
Cano-Sarmiento, C.; Téllez-Medina, D.I.; Viveros-Contreras, R.; Cornejo-Mazón, M.; Figueroa-Hernández, C.Y.; García-Armenta, E.; Alamilla-Beltrán, L.; García, H.S.; Gutiérrez-López, G.F. Zeta Potential of Food Matrices. Food Eng. Rev. 2018, 10, 113–138.
Line 224 - Please avoid using the term had, please use exhibited/demonstrated instead
Line 404 - What shear rate should the reader look at? The authors discussed a general tendency? The authors should guide the readers’ eyes to the specified point.
I mean: For instance, at the shear rate of 50 s-1, the highest viscosity was observed for the…
Line 434 – too many commas
Foaming capacity and foaming stability paragraph - please cite the recent paper on the effect of pH on protein foaming, because the citations that you have provided are little outdated
Nastaj, M.; SoÅ‚owiej, B.G. Effect of various pH values on foaming properties of whey protein preparations. Int. J. Dairy Technol. 2020, 73, 683–694.
Line 475 - …, which were conductive to the adsorption
Conclusions
Please emphasize what applications the authors see for the resulting microparticles.
of course, in food, but speaking of samples obtained at pH 9 and PH 11, I just don't know how they taste and what effect microparticles obtained in such an alkaline environment may have on the body. have you wondered? I think this is worth considering and commenting on in the paper. If not food, maybe non-food industrial foams then?
References
please adjust references to MDPI standards
line 527- ?????
Good luck with the corrections!
Author Response
We thank the reviewer for their complimentary comments and contribution to improvements of this manuscript. We have carefully revised, supplemented, and explained in accordance with the reviewers' comments and questions, and checked and checked the language of the full text. The annex lists the responses to the detailed comments.
